# Development of Human Toxo IgG ELISA Kit, and False-Positivity of Latex Agglutination Test for the Diagnosis of Toxoplasmosis

**DOI:** 10.3390/pathogens10091111

**Published:** 2021-08-31

**Authors:** Haroon Akbar, Muhammad Zubair Shabbir, Ubaid Ullah, Muhammad Imran Rashid

**Affiliations:** 1Department of Parasitology, University of Veterinary and Animal Sciences, Lahore 54000, Pakistan; sarfrazvet@gmail.com (S.-u.-R.); drharoonakbar@uvas.edu.pk (H.A.); drubaid786@gmail.com (U.U.); 2Institute of Microbiology, University of Veterinary and Animal Sciences, Lahore 54000, Pakistan; shabbirmz@uvas.edu.pk

**Keywords:** toxoplasma, toxoplasmosis, IgG ELISA, Western blot, SAG1

## Abstract

*Toxoplasma gondii* is an intracellular zoonotic parasite that causes infection in a wide range of warm-blooded animals and humans. The main aim of this study was to assess the diagnostic value of the recombinant SAG1 antigen (rSAG1) for *T. gondii*-IgG screening through the Human Toxo IgG ELISA Kit (K). The rSAG1 was expressed in *E. coli* (DE3), and it was purified through metal-affinity chromatography. The rSAG1 was confirmed by immunoblotting, and it had a band on 35 kDa. Total of 400 human sera were tested by LAT and K. One hundred and twenty-two (30.5%) sera were found positive by LAT and eighty-nine (22.25%) sera were found positive by K. Out of 400 samples, 80 were selected to evaluate the performance of K through commercial *Toxoplasma gondii* IgG ELISA Kit (C). Out of 80 human sera, 55 (68.75%) were found positive, 25 (31.25%) were found negative by K and C, respectively. The cut-off value for K was 0.398 and it was calculated through the receiver operator characteristic curve. The ELISA plates were coated at optimized concentration of rSAG1 = 0.125 µg/mL, and the test was performed by diluting the sera at 1:50. The sensitivity and specificity of K were observed to be 98.5% and 100%, respectively. The six sera (K^−^L^+^) were found positive through LAT and these human sera were later evaluated by Western blot analysis. These sera did not produce a band equivalent to 35 kDa on WB analysis thus, LAT produced false-positive results.

## 1. Introduction

*Toxoplasma gondii* is an intracellular zoonotic parasite that causes toxoplasmosis globally in warm-blooded animals and humans [1,2]. In healthy individuals, *T. gondii* infection is usually asymptomatic whereas, it causes life-threatening problems in immunocompromised subjects. Moreover, in pregnant women, it causes serious problems due to the transplacental transmission from mother to fetus and thus resulted in miscarriage, permanent neurologic damage, premature birth, and visual disorders. Furthermore, immunocompromised (due to AIDS like other immunosuppressed diseases) individuals have more possibilities to acquire the toxoplasmic encephalitis and it can be life-threatening [3,4]. The methods of transmission of *T. gondii* in humans is through eating of food and drinking of water contaminated with oocysts of *T. gondii* or the consumption of raw or undercooked meat having tissue cysts containing bradyzoites [1,4,5]. The prevalence of toxoplasmosis fluctuates among the individuals and different age groups [5]. In the United States, one of the most common cause of food-borne deaths are estimated due to toxoplasmosis [6]. An annual cost of around $7.7 billion on illness due to toxoplasmosis is reported in the USA [7]. In Pakistan, the overall prevalence rate of this disease in humans is from 12% to 28% [8,9,10]. The prevalence of toxoplasma infection is higher in pregnant women in Khyber Pakhtunkhwa; 38%, Azad Jammu and Kashmir; 48%, and Punjab; 63% [10].We have already identified *T. gondii* from fecal samples of cats at Pet Centre of UVAS (Lahore, Pakistan), based on SAG2 sequence, it was an atypical strain (UVAS-Toxo-1, UVAS-Toxo-3, UVAS-Toxo-6) of *T. gondii* in Pakistan [11] whereas, the atypical strain of *T. gondii* is also present in other countries including SouthAmerica, Central America, North America, the Caribbean, and Africa [12]. This genetic diversity of *T. gondii* is due to incidence of genomic recombination, distinct population structures, and intercontinental regional diversity [13].

There are several methods for the detection of toxoplasmosis. These methods include genetic analysis of *T. gondii*, proteomic analysis, and screening of antibodies [14,15]. The commercially prepared kits of *T. gondii* are mostly prepared by using whole parasite’s antigens out of tachyzoites [15,16]. Moreover, it is vulnerable to carry live parasites. These antigenic problems can be resolved through recombinant DNA technology to produce recombinant protein [17].

In recent few years, several recombinant proteins were used for the detection of specific IgG against *T. gondii* parasite in the animal and human hosts. These diagnostic antigens are based on surface antigens like SAG1, SAG2, SAG3 [18,19,20,21,22,23,24], dense granule antigens like GRA1, GRA2, GRA4, GRA5, GRA6, GRA7 [24,25,26,27,28,29,30], microneme antigens like MIC2, MIC3, MIC4, MIC5 [14,31,32], matrix antigens like MAG1 [33], and rhoptry antigens like ROP1 and ROP2 [24,34,35]. Different types of studies reported the successful diagnostic importance of recombinant antigens for the identification of anti-*T. gondii* antibodies. The recombinant antigens individually or in combination forms have been reviewed for describing their diagnostic sensitivity [32]. We had previously used SAG1 for ELISA in infected mouse with the support of the project of Grand Challenges Canada (Grant # S4_0266–01) [36].The different cloned genes of *T. gondii* indicate that SAG1has proved as a good candidate for serodiagnosis of infection of *T. gondii* [37,38]. It is a highly conserved, and immunodominant antigen for identification of different strains of *T. gondii* [39,40,41]. It is used for recognition of acute, and chronic phases of toxoplasmosis [15,42]. In serology, ELISA is more satisfactory for identification of *T. gondii* infection as compared to the modified agglutination test (MAT) [43]. In our study, the Human Toxo IgG ELISA Kit (K) was developed with rSAG1 and the diagnostic value for specificity and sensitivity was determined by comparing with LAT and commercially available kit named *Toxoplasma gondii* IgG ELISA (C) Kit. The samples found positive with LAT and negative with K, were analyzed with WB. The results of WB showed the false positivity of the sera with LAT as compared to K. The objective of development of local diagnostic ELISA kit was to diagnose toxoplasmosis in Pakistan, and to improve the sensitivity and specificityas compared to commercial diagnostic ELISA kits. The main advantages of our local diagnostic ELISA kit are; firstly, it will be easily accessible, secondly, it might be economical because of small quantity of recombinant protein of *T. gondii* was used in optimization of ELISA.

## 2. Results

### 2.1. Transformation and Restriction Analysis of pET28a-SAG1

The transformed colonies of BL21 with pET28a-SAG1 were clear and white as shown in Figure 1A. Two fragments were observed with double digestion through XhoI and NheI at 957 bp and 5369 bp, respectively, as shown in Figure 1B.

### 2.2. Induction and Expression of rSAG1

The induced cultures had bands at 35 kDa while no bands were observed at 35 kDa in uninduced cultures, and BL21 cultures as shown in Figure 2A. The purified rSAG1 was identified through WB by using Anti-6x His Antibodies and 35 kDa band [36] was detected as shown in Figure 2B.

### 2.3. Performance of LAT (L) and Human Toxo IgG ELISA Kit (K)

A total of 400 samples were tested by L and K. In total, 30.50% were positive by L and 22.25% were positive by K, 69.50% were negative by L and 77.75% were negative by K, respectively, as described in Table 1.

The diagnostic performance of L and K was assessed through positive and negative sera detected by L and K. The percentage of sensitivity and specificity were 72.95% and 89.71%, respectively. The kappa value was calculated as 0.789. The sensitivity of LAT was found to be too low. The Chi-square (χ^2^) value was 27.15 and was found significant *p* < 0.05 as shown in Table 2.

### 2.4. Evaluation of Sera through Toxoplasma gondii IgG ELISA Kit (C)

A total of 80 samples were selected for commercial ELISA to check the association of K and C on the criteria of positive and negative through L and K. Further, 100% sera were observed positive by C for 55 samples while for 25 sera, 100% of samples were found to be negative by C as described in Table 3.

### 2.5. Determination of Sensitivity, Specificity, and Cut-Off Value of K

By comparing with C, sensitivity and specificity of K were determined as 98.5% and 100%, respectively and kappa value was calculated as 0.971 which was found highly significant (*p* < 0.001) as shown in Table 4.

The sera were tested by K, positive and negative sera were decided on the basis of cut-off value. The positive and negative sera for toxoplasmosis were checked by K which was helpful to determine the cut-off value. The cut-off value was 0.398. The sensitivity was 98.5% and specificity was 100% as shown in Figure 3 [44].

The cut-off value was 0.398 on which the positive and negative sera were decided. The value of area under curve (AUC) was highly significant (*p* < 0.001). The AUC was 1.000. ROC analysis depicted the comparisons of cut-off values with respect to sensitivity and specificity of K. Similarly, the AUC explained the overall performance of diagnostic test. AUC = 0.5 is not good and non-informative for the diagnostic test, AUC = between 0.5 and 0.7 depicts the less accurate value for the diagnostic test, AUC = between 0.7 and 0.9 indicates the moderately accurate, AUC = between 0.9 and 1 indicates the highly accurate value for the diagnostic test while AUC = 1.000 indicates the perfect test, and the diagnostic test is perfect for the diagnosis of positive and negative sera [45]. Subsequently, AUC = 1.000 of K indicated that this test was perfect test for the diagnosis of positive and negative sera of toxoplasmosis.

### 2.6. False-Positivity of L through WB

The positive human serum sample number 53 as a control has a band at 35 kDa by WB analysis, and it was also positive by K and C. The six negative human sera; 22, 131, 137, 288, 261, 339 have no band at 35 kDa by WB analysis, whereas these 6 sera were positive by L and negative by K and C for *T. gondii* infection as shown in Figure 4.

## 3. Discussion

In the current study, our Human Toxo IgG ELISA Kit was developed with rSAG1 of *T. gondii*.The sequence of SAG1 was 957bp, it was used for the production rSAG1. For that rSAG1 was produced in prokaryotic expression system; BL21 having transformed with pET28a-SAG1 by heat shock methodthe induction of rSAG1 was done with 1 mM IPTG. The 2× Laemmli buffer was used in SDS-PAGE while protein estimation was performed by a BCA kit method. The size of rSAG1 was 35 kDa. The rSAG1 was coated on ELISA plate at concentration at 0.125 µg/mL. The 400 human sera were evaluated through a K and latex agglutination test for the detection of anti-*T. gondii* IgG antibodies. Out of 400, 80 samples were selected for their screening through commercially available ELISA kit. We found 98.5% sensitivity, 100% specificity, and a 0.398 cut-off value of our ELISA kit (K) in comparison to the *Toxoplasma gondii* IgG ELISA kit (C). By using western blot, we found false positivity of LAT in 6 samples.

In our previous study, we coated ELISA plate at 5 µg/mL to screen the antibodies of vaccinated mice immunize intranasally with SAG1 nanoparticles [36]. However, in the current study, the ELISA plate was coated at 0.125 µg/mL to screen the antibodies in human infected with *T. gondii.*

The SAG1 protein has been extensively used as an excellent and widely explored antigenfor the detection of antibodies against *T. gondii* [18,32]. It was noted that rSAG1 was the immunodominant and good candidate for the identification of *T. gondii* infection. SAG1 is more dominant than other proteins like SAG2, SAG3, GRA1, GRA2, GRA3, MIC2, MIC3, ROP1 of *T. gondii* because of the interactions of multiple disulfide bonds and C-terminal hydrophobic regions [37,46]. Native SAG1 protein has six intramolecular cysteine bridges that makes it immunodominant because of having immunologically confirmatory epitopes [38,47].

For the optimization of our ELISA kit, diagnostic value of rSAG1 was evaluated in decreasing concentrations i-e 1 µg/mL, 0.5 µg/mL, 0.25 µg/mL, and then 0.125 µg/mL. Finally, we have reached at 0.125 µg/mL concentration for our optimal assay. In a study by Nouha et al. (2013), 867 bp of DNA sequence of *T. gondii* was cloned into plasmid pET22b (+) encoding for protein comprising of amino acids from 47 to 336 of SAG1. The plate was coated at 100 µg/mL for ELISA to diagnose toxoplasmosis the size of rSAG1 was 38 kDa [44]. ELikira et al. (2001), cloned SAG1 into pGEX-4T-3 and expressed in BL21. The plate was coated with rSAG1 of *T. gondii* at 96 µg/mL for immunological assay. The positive rate of the ELISA for toxoplasmosis was 20.5%. The sensitivity and specificity were low as compared to BAG1 [48]. Selseleh et al. (2012) developed an ELISA with rSAG1 at 5 µg/mL for the detection of anti-*T. gondii* antibodies in human sera. SAG1 was cloned into pET28a.The induction of rSAG1 was done through IPTG. The sensitivity and specificity of developed IgG ELISA were 93%, and 95%, respectively, as comparison was done with commercial ELISA [49]. Kimbita et al. (2001) coated the plate with a concentration of rSAG1 at 2 µg/mL on the surface of 96-well plate for ELISA for the detection of toxoplasmosis, whereas positive rate was 20.7% and negative rate was 79.3% as compared to Western blot [50]. Madhurendra et al. (2018) coated rSAG1 at 2 µg/100 µL of carbonate-bicarbonate buffer per well for ELISA to diagnose toxoplasmosis. The sensitivity and specificity of indirect ELISA were 92.66%, and 90.67%, respectively, when it was compared to IFAT [51]. Si et al. (2004) developed a rSAG1-ELISA and plate was coated at 5 µg/mL The positive rate and negative rate were 81% and 95%, respectively as compared to imported IgG-ELISA kit [52]. The differences in the level of detection of rSAG1 in the human sera might be due to the differences in different expression system and variable gene fragments [38].

*Toxoplasma gondii* IgG ELISA Kit is a yardstick and reliable serological method for detection of IgG antibodies of T. gondii. It has sensitivity of 98.3%, specificity of 99.2%, and accuracy of 98.9%. The commercial kit is used for the evaluation of serological status of human to diagnose *T. gondii* infection. The positive and negative controls were obtained from World Health Organization (WHO) and these positive and negative sera were used in C. In a study by Lin et al. (1988), they used the WHO serum as a standard for the development of ELISA for the detection of antibodies against *T. gondii* [53]. Its protocol is only 90 min. The performance of K was analyzed with C.

The cut-off value and area under curve (AUC) of K was determined by ROC. The ROC is used to find out the cut-off value of ELISA for the detection of positive and negative sera for anti-*T. gondii* antibodies. It is also used to determine the efficacy of diagnostic kits for detection of antibodies [44,45]. In the study of K development, the cut-off value was 0.398, AUC for K was 1.000, and it was highly significant (*p* < 0.001). The AUC explains the overall output of diagnostic test. AUC = 0.5 is not good for the diagnostic test, AUC = between 0.5 and 0.7 depicts the less accurate value, AUC = between 0.7 and 0.9 is moderately accurate, AUC = between 0.9 and 1 depicts the highly accurate value while AUC = 1.000 indicates the perfect test [45]. Mangili et al. (2009) used ROC curve to find the AUC for the evaluation and performance of in-house ELISA for diagnosis of toxoplasmosis from serum. The AUC of in-house ELISA was 0.935 when it was comparison to commercial ELISA for diagnosis of toxoplasmosis [54]. Glor et al. (2013) calculated the AUC for the performance of ELISA for diagnosis of toxoplasmosis in comparison to IFAT through using ROC curve. The AUC for ELISA was 0.991 relative to IFAT [55].

In the current study, we found 6 serum samples (22,131,137,261,288,339) which were positive by LAT and negative by both K and C. We analyzed these samples with WB. LAT is considered the initial screening test for toxoplasmosis [56], but not reliable and accurate method like rSAG1-based ELISA for the identification of toxoplasmosis [57]. It was indicated that LAT is not confirmatory test for toxoplasmosis. WB is useful serological tool to determine the presence and absence of *T. gondii* antibodies, and it can reliably be used for the presence or absence of antibodies [58]. These 6 sera were found negative on WB analysis. Thus, LAT showed false positive results on these samples. The false-positivity of LAT had already been described by other researchers. One scientific group described that the sera have high concentration of hepatitis B virus “e” antigen can give false positive reactions by using LAT. The reason might be due to cross reactivity of BSA used in preparation of latex particles which is also used in the “e” antigen. Even this reactivity was unaffected by the treatment of 2-mercaptoethanol of the sera [59]. Similarly, another scientific group found that some sera showed false-positive results by LAT [59]. It was reported that a LAT for identification of toxoplasmosis (Toxo Reagent, Mast Diagnostics), gave false-positive results with serum samples of the patients who have transplanted heart, having cytomegalovirus (CMV). The virus produces non-toxoplasma specific antibodies that are associated with albumin and toxoplasma antigen, but the antibodies are not specific to the antigen, whereas false-positive results are produced due to polarity on antigen [60]. The LAT is a test for the identification of suspected toxoplasmosis, but the false positive reactions occur due to false reaction of antibodies to toxoplasma latex antigen.

## 4. Materials and Methods

### 4.1. Ethics Statement

The study was conducted following the guidelines of the Human Ethics Committee of UVAS, Lahore, Pakistan (No. 056/IRC/BMR, dated: 31 July 2019).

### 4.2. Sketch of the Experiments

The flow diagram of our experiments conducted is shown in Figure 5.

### 4.3. Collection of Human Sera

A total of 400 human sera were collected from the patients of different ages. The sera were collected from Central Diagnostic Laboratory, Mayo Hospital, Lahore.

### 4.4. Screening of Sera throughthe Latex Agglutination Test (LAT)

The LAT was performed as described by protocol [61] (Toxo-Latex, Linear Chemicals, Spain, Latex Kit Lot # 20072712). The slide agglutination in suspension of latex particles showed the complex of antibodies of unknown samples with antigen of *T. gondii*. The presence or absence of agglutination depicts the positivity or negativity of samples for anti-toxoplasma antibodies. Briefly, each serum sample was diluted to 1:4 with 0.1 M PBS. Twenty five µL of diluted serum was mixed on the blackish region of the glass slide. Positive and negative control sera were mixed similarly with the help of spatula. The slide was rotated by a rocker for 3 to 5 min, and the agglutination was observed by gross visualization. The observed agglutination was considered positive or negative after comparison with the positive and negative controls.

### 4.5. Expression and Purification of rSAG1

We had already amplified SAG1 from positive *T. gondii* oocyst from the cat feces. SAG1 was cloned at XhoI/Nhe1 in pET-28a as previously described [36,62]. Competent cells were prepared as described by the protocol [63]. Briefly, the *E. coli* BL21 (DE3) strain was grown in the Luria Bertani (LB) broth and was incubated at 37 °C, 0.66× *g* for 18–24 h. After incubation, *E. coli* culture was diluted 100 folds in fresh LB broth and was incubated at 37 °C, 0.66× *g* for 4–5 h until the value of optical density (OD) reached 0.5 at 600 nm. After attaining the desired OD, the culture was cooled down to 0 °C by pouring it into 50 mL of sterile, pre-chilled tube containing CaCl_2_ and it was centrifuged at 1431× *g* 4 °C for 10 min. The supernatant was discarded and 30 mL of chilled 100 mM of CaCl_2_ was added_._ The mixture was again centrifuged at the same conditions. Later, 10 mL of 15% (*v/v*) glycerol (Thermo Fisher scientific, Life Technologies Co. Ltd., Zhongzheng, Taipei city, Taiwan, Cat# 15514-011) was added in the pellet and it was re-suspended again. The treated cells were kept on ice and then these were stored in 1.5 mL of cryo-preservative vials in a volume of 500 µL. The competent cells were stored at −196 °C until further use.

Naeem et al. (2018) have explained the protocol of the expression of rSAG1. Firstly, transformation of pET28a-SAG1 in competent cells was done with heat-shock therapy in water bath at 42 °C for 90 s. An aliquot of 100 µL of competent cells was mixed with 1 µL of the plasmid. Immediately after heat shock, the tube was shifted on ice for 30 min. Then, 900 µL of warmed LB broth was added into the mixture and then it was incubated at 37 °C, 0.66× *g* for 1 h. The 100 µL of the aliquot was spread on LB agar plates containing kanamycin 1 mg/mL. The plates were incubated at 37 °C for 24 h. After 24 h, the plates were observed for transformed colonies. The confirmation of transformation was performed by restriction analysis by using restriction enzymes Nhe1 and Xho1. Briefly, 3 U of enzymes and the compatible 10X buffer were used to digest 1 µg of plasmid at 37 °C for 1 h. The BL21 (DE3) strain colonies were selected from LB agar plate having kanamycin 1 mg/mL. The colonies were inoculated in 10 mL of fresh LB broth and were incubated at 37 °C, 0.66× *g* for 24 h until the OD_600_ reached to 2.0. After attaining the desired concentration, the culture was diluted 100 folds in 50 mL of fresh LB broth and again it was incubated at 37 °C, 0.66× *g* for 4 h until the OD_600_ reached to 0.5. After that, 1.0 mM isopropyl-β-D-thiogalactopyranoside (IPTG) (Bio-World, Dublin, OH, USA, Cat# 715052) was added in the culture and again it was incubated at 37 °C, 0.66× *g* for 6 h. Subsequently, the induced cells were lysed with the help of 5 × loading buffer (containing 10% SDS, 100 mM DTT, 50% glycerol, 0.05% bromophenol blue, 0.313 M Tris–HCl of pH 6.8). After lysis, the induced and un-induced cells were loaded on 12%sodium dodecyl sulfate-polyacrylamide gel electrophoresis (SDS-PAGE) for further purification of rSAG1 protein of *T. gondii.* HisPur nickel-nitrilotriacetic acid (Ni-NTA) affinity columns (Thermo Fisher scientific, Life Technologies Co. Ltd., Zhongzheng, Taipei city, Taiwan, Cat # 88228) was used for the chromatographic purification of rSAG1 protein.

The induced cells were lysed for purification of protein by standard protocol [64]. Briefly, the cells of pellet were re-suspended in lysis buffer (20 mM of NaH_2_PO_4,_ 8 M of Urea, 500 mM of NaCl (pH 8.0). A sonic pressure wave was generated by ultrasonic vibrations (15–20 kHz) and the sonic pressure wave was used for lysis of cells. These waves were in the form of pulses [65]. After adding lysis buffer, the suspension was sonicated for 30 pulses for 15 times at 1 min interval. During sonication, the tube containing lysate was put in an ice container. The rSAG1 was quantified by bicinchoninic acid (BCA) assay kit (G-Biosciences, Saint Louis, MO, USA, Cat# 786-570). The purified rSAG1 was verified through WB by immunoblotting with Anti-6× His Tag Antibodies (ThermoFisher, Houston, TX, USA, Cat# MA1-21315).

### 4.6. Screening of Sera through Human Toxo IgG ELISA Kit (K)

The rSAG1 was coated on 96-well flat bottom, polystyrene plate (JET BioFil, Hong Kong, China, Code# TCP011096) at 0.125 µg/mL in coating buffer (50 mM Na_2_CO_3_), and it was incubated at 4 °C overnight. The washing of plate was done 5 times with the washing buffer (0.001 M PBS/0.05% Tween-20) at 300 µL/well. The plate was saturated with 4% BSA in 0.01 MPBS with 200 µL/well and it was incubated at 37 °C for 2 h. Again, the plate was washed 5 times with the washing buffer. Four hundred human serum samples were screened through K. The positive and negative control sera were dispensed in duplicate wells. Two wells were put as blanks in the plate. The plate was incubated at 37 °C for 2 h. After washing 5 times, anti-human secondary antibodies (Invitrogen ThermoFisher, Houston, TX, USA, Cat # 31310) conjugated with alkaline phosphatase (AP) were dispensed with 1:5000 dilution at 100 µL/well and the plate was incubated at 37 °C for 2 h. After washing 5 times again, the substrate *p*-nitrophenyl phosphate (pNPP) (Thermo Scientific, USA, Ref # 34045) was added with 1 mg/mL of DAE substrate buffer (ThermoFisher, USA, Cat # 34064) at 100 µL/well. The plate was incubated at 37 °C and the reaction was stopped after 15 min by adding stop solution (1 M NaOH) at 100 µL/well. The OD was taken at 405 nm by Microplate ELISA reader (ELX-800, Tennessee, BioTek, Winooski, VT, USA).

### 4.7. Screening of Sera through Toxoplasma gondii IgG ELISA Kit (C)

Out of 400 sera, 80 were selected for C on the basis of positive and negative sera through K and L. The detail of selection is shown in Table 5.

*Toxoplasma gondii* IgG ELISA Kit (C) (Abnova antibody innovation, New Taipei city, Taiwan, Cat# K0225) was purchased from ABNOVA Company. The pre-coated plate with rSAG1 was added with diluted sera. The unbound antibodies were washed away by the washing solution. Conjugate attached with HRP was added in wells that were attached to primary antibodies. The antibody-antigen complex was revealed by adding tetramethylbenzidine (TMB) solution. The reaction was stopped by adding distilled water. The results were compared with calibrators and controls. The 80 human sera were dispensed into 96-well plate. The negative, positive and calibrators were added in duplicate with 1:40 dilution. A well was used as a blank. The plate was incubated at 37 °C for 30 min, and it was washed 5 times with washing solution. Then, 100 µL of enzyme conjugate was added in each well. The plate was incubated at 37 °C for 30 min, and it was washed again 5 times with washing solution. The 100 µL TMB substrate was added in each well. The plate was incubated at 37 °C for 15 min. Then, 100 µL of stop solution (1 N HCl) was added in each well. After 15 min of addition of substrate, the reading of OD was taken at 450 nm.

### 4.8. Screening of Sera through Western Blot (WB) Technique

Out of 400 sera, 6 sera were selected that were negative by K and positive by L. The SDS-PAGE was performed to separate the purified rSAG1 on 12% polyacrylamide gel and the standard protein marker (SDS–PAGE standards, Bio-Helix Co. Ltd. (GeneDirex), New Taipei city, Taiwan, Cat# PM007-0500) was run along with samples for comparison. The proteins on gel were transferred to nitrocellulose membrane (NCM) (Guangzhou World Trade Center Complex, Guangzhou, China, Cat#LC2000), and it was carried out in Trans-Blot Turbo Machine (Trans-Blot Turbo Transfer System, Bio-Rad) at 25 volts, and 1.3 mA for 7 min. The NCM was washed 3 times with Tris-Buffered Saline, 0.1% Tween buffer (TBST) (20 mM Tris HCl, 0.1% Tween 20 and 150 mM NaCl) 3 times for 15 min. The membrane was blocked by 5% skimmed milk-TBST at 4 °C overnight. Then, NCM was washed with TBST buffer 3 times for 15 min each at room temperature. The NCM was incubated with the serum of *T. gondii*-infected human with 1:50 dilution in 5% skimmed milk-TBST for 2 h. The immunoblot was revealed by using anti-human IgG (Invitrogen ThermoFisher, Houston, TX, USA, Cat # 31310) conjugated with AP with 1:500 dilution conjugated with AP. The membrane was washed with TBST for 3 times for 15 min each. The 5-bromo-4-chloro-3-indolyl phosphate/nitroblue tetrazolium (BCIP/NBT) (Sigma-Aldrich, Roche Diagnostic GmbH, Mannheim, Germany, Cat # 11442074001) was used for the detection of protein bands on the immunoblot. Distilled water was added to stop the reaction. A total of seven human sera were verified by WB analysis. One human serum sample was used as a control which was found positive by both K and C.

### 4.9. Determining the Percentages of Specificity and Sensitivity and Cut-Off Value of K

Out of the 80, 55 positive and 25 negative sera were evaluated by C. An interactive diagram of the receiver operating characteristic (ROC) curve was used to determine the sensitivity and specificity of K [44]. The ROC curve was also used to determine the cut-off value [44].

### 4.10. Development of Human Toxo IgG ELISA Kit (K)

The levels of rSAG1, levels of serum dilutions, and incubation times were optimized with repetition of tests until the optimized results were obtained. For optimization of rSAG1 protein, coating of wells with rSAG1 was done at decreasing concentrations of rSAG1; 1 µg/mL, 0.5 µg/mL, 0.25 µg/mL, and 0.125 µg/mL. Subsequently, different dilutions of sera; 1:200, 1:100, 1:50 were also used until the dilutions were optimized. Reading of OD value was also taken at different time intervals; 5 min, 10 min, 15 min, 20 min and 30 min, until we reached an optimal time of reading after adding the substrate. Thus, the coating concentration of rSAG1 for K was 0.125 µg/mL, and the dilution of human sera was optimized at 1:50. Subsequently, the incubation time for sera and antibodies was optimized at 37 °C for 2 h. After addition of substrate, the reading of OD time was optimized at 15 min. These optimized conditions showed good diagnostic test values of K.

### 4.11. Statistical Analysis

The MedCalc statistical software (version 11.4.4.0) was used to compare performance of K, L, and C by calculating kappa value. The ROC curve was used to find out the cut-off value of K [44].

## 5. Conclusions

The K for toxoplasmosis detects anti-toxoplasma antibodies in human sera at 98.5% sensitivity and 100% specificity. The results of LAT showed false-positivity. These false-positive results were also tested through WB. The results of this study concluded that K is more accurate, reliable and a potential immunodiagnostic test than LAT for the diagnosis of *T. gondii* infection in humans.

## Figures and Tables

**Figure 1 pathogens-10-01111-f001:**
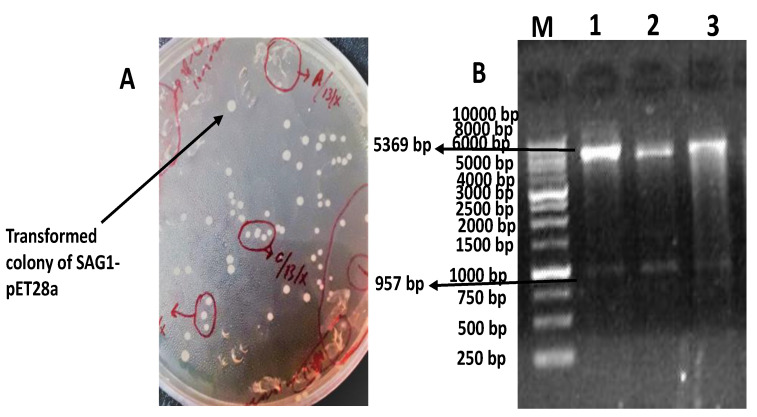
Describing the transformation of pET28a-SAG1 and restriction analysis. The colonies’ transformed bacteria are clear and spherical (**A**). The Lane M shows 1 kb DNA ladder (Thermo Fisher Scientific Baltics, Graiciuno, Vilnius, Lithuania, Cat# SM0312), Lanes 1, 2, and 3 show different samples showing two fragments of 957 bp and 5369 bp on double digestion with XhoI and NheI (**B**).

**Figure 2 pathogens-10-01111-f002:**
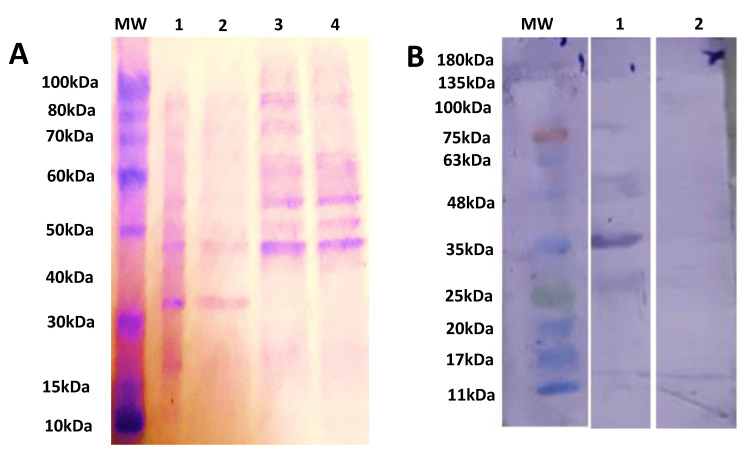
Describing the protein separation and identification bySDS-PAGE and WB. The SDS-PAGE analysis of expression of rSAG1 using 12% acrylamide gel. Lane M shows protein marker (ThermoFisher Scientific, Life Technologies Co. Ltd., Zhongzheng, Taiwan, Cat# 10747-012), Lane 1 shows expression after 7 h of induction, Lane 2 shows expression after 4 h of induction, Lane 3 shows expression without induction, Lane 4 shows expression of competent BL21 culture after 7 h of induction (**A**). WB analysis of the rSAG1 using an Anti-6x His Tag Antibodies (ThermoFisher Scientific, Life Technologies Co. Ltd., Zhongzheng, Taipei city, Taiwan, Cat# MA1-21315). Lane M shows the protein marker—(Bio-Helix Co. Ltd. (GeneDirex), New Taipei city, Taiwan, Cat# PM007-0500), Lane 1 shows purified rSAG1 protein, Lane 2 shows induced control culture of cells without insertion of SAG1 (**B**).

**Figure 3 pathogens-10-01111-f003:**
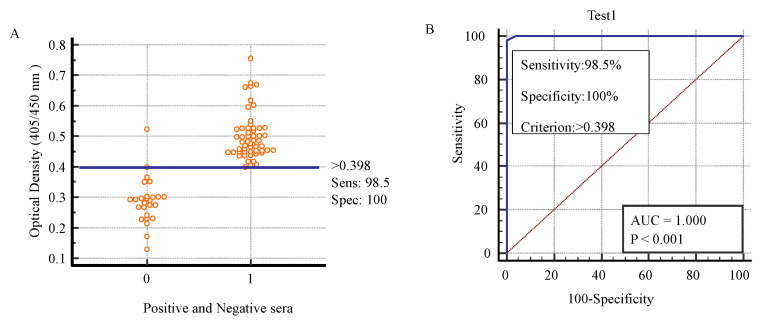
Describing the sensitivity, specificity, and cut-off value of K according to the toxoplasma-positive and -negative sera by K to determine sensitivity and specificity. Sensitivity (Sens), specificity (Spec). (**A**). ROC curve for K applied to positive versus negative individuals identified by C to determine cut-off value (**B**).

**Figure 4 pathogens-10-01111-f004:**
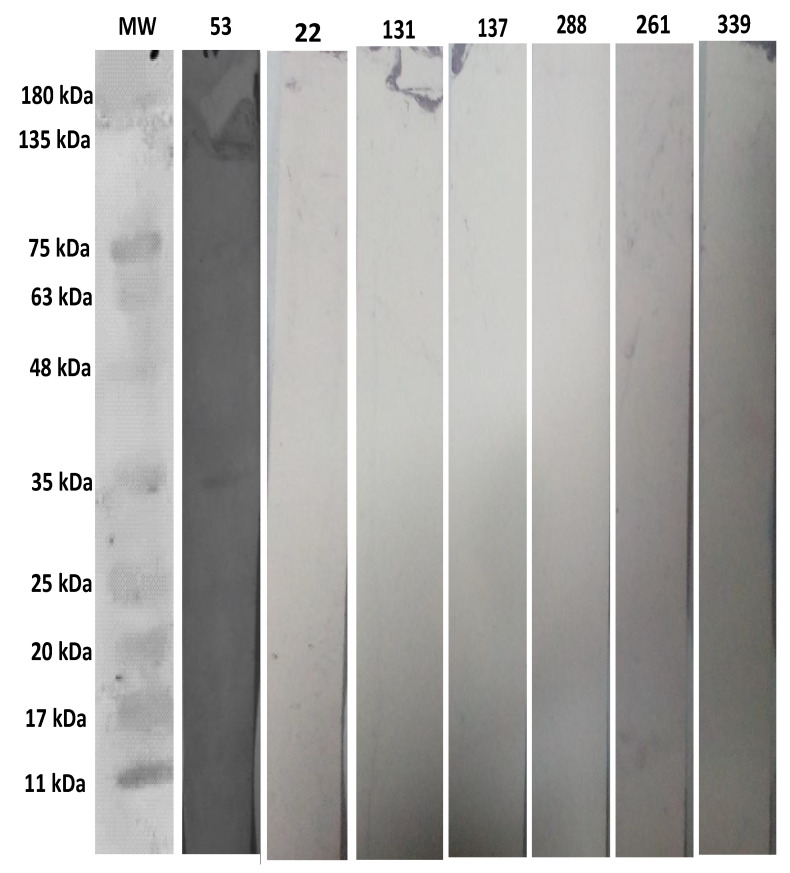
Describing the WB analysis of the negative sera of L using an anti-human IgG (Invitrogen ThermoFisher, Houston, TX, USA, Cat# 31310) conjugated with AP. Lane 1 shows protein marker (Bio-Helix Co. Ltd. (GeneDirex), New Taipei city, Taiwan, Cat#PM007-0500), Lane 2 shows positive human serum sample number 53, Lanes 3, 4, 5, 6, 7, and 8 show negative serum samples 22, 131, 137, 288, 261, and 339, respectively.

**Figure 5 pathogens-10-01111-f005:**
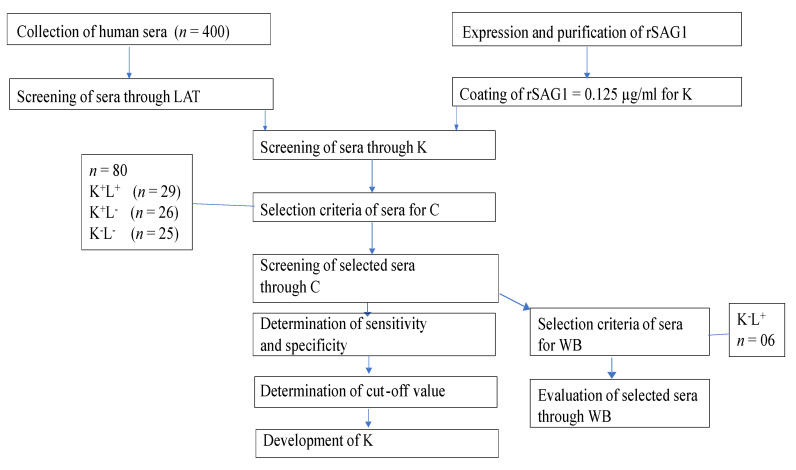
Flow chart of experimental plan. Human Toxo IgG ELISA Kit (K), *Toxoplasma gondii* IgG ELISA Kit (C), Western blot (WB), latex agglutination test (LAT), LAT (L), positive (+), negative (−).

**Table 1 pathogens-10-01111-t001:** Summary of results from L and K for the detection of anti-*T. gondii* IgG in 400 sera.

Results	L	K	Description		
Positive	122 (30.50%)	89 (22.25%)	K^+^L^+^ (33 sera) (8.25%) (a)	K^−^L^+^ (91 sera) (22.75%) (b)	124 (a + b)
Negative	278 (69.50%)	311 (77.75%)	K^−^L^−^ (220 sera) (55%) (c)	K^+^L^−^ (56 sera) (14%) (d)	276 (c + d)
**Total**	400	400			a + b + c + d

Numbers of sera K^+^L^+^ (a), numbers of sera K^−^L^+^ (b), numbers of sera K^−^L^−^ (c), numbers of sera K^+^L^−^ (d).

**Table 2 pathogens-10-01111-t002:** Assessment of K and L for the detection of anti-*T. gondii* IgG in 400 sera. Positive predictive value (PPV), negative predictive value (NPV).

Results	rSAG1 (%)
Sensitivity	72.95
Specificity	89.71
PPV	72.95
NPV	89.71
validity	81.33
Kappa (95% CI)	0.789 (0.64–0.83)
Standard error	0.034
χ^2^ = 27.15	
*p* < 0.05	

**Table 3 pathogens-10-01111-t003:** Showing the results of 80 human sera tested through C for the detection of anti-*T. gondii* IgG.

Results	Selected Sera	Screening Through K	Screening Through C
K^+^L^+^	29	29	29
K^+^L^−^	26	26	26
K^−^L^−^	25	25	25
Total	80	80	80

**Table 4 pathogens-10-01111-t004:** Assessment of K and C for the detection of anti-*T. gondii* IgG in 80 sera. Positive predictive value (PPV), negative predictive value (NPV), area under curve (AUC).

Results	rSAG1 (%)
Sensitivity	98.5
Specificity	100
PPV	98.21
NPV	100
validity	98.075
Relative agreement	81.45
Kappa (95% CI)	0.971 (0.64–0.83)
Standard error	0.028
AUC	1.000
Accuracy	98.77
χ^2^ = 8.02	
*p* < 0.05	

**Table 5 pathogens-10-01111-t005:** Selection of sera for C. LAT (L), Human Toxo IgG ELISA Kit.

Results	Selection Through C
K^+^L^+^	29
K^+^L^−^	26
K^−^L^−^	25
Total	80

## Data Availability

The data will be available to corresponding author on request.

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
