# Peer review of "Development of Human Toxo IgG ELISA Kit, and False-Positivity of Latex Agglutination Test for the Diagnosis of Toxoplasmosis"

_pathogens, 2021, doi:10.3390/pathogens10091111_

Round 1
Reviewer 1 Report
Sarfraz-ur-Rahman and cols. presented a well sustained research, introduction provides the required background in order to understand the entire article, nevertheless I have a main observation:
1. Authors must improve Figures 2A, 3 and 5
Author Response
Reviewer # 1: Sarfraz-ur-Rahman and cols. presented a well sustained research, introduction provides the required background in order to understand the entire article, nevertheless I have a main observation:
Comment 1: Authors must improve Figures 2A, 3 and 5
Response: The Figures 2A, 3 and 5 are improved at the manuscript page no. 3,4 and 8 and line numbers 72,83, and 152, respectively.

Reviewer 2 Report
In Figure 5 I cannot see the positive bands in lane 2
Author Response
Reviewer # 2: In Figure 5 I cannot see the positive bands in lane 2
Response: Now the Figure 5 is improved and band in the line 2 is prominent. The improved figure is at page no. 8 and line number 152 in the manuscript.

Reviewer 3 Report
This manuscript focuses on an important topic in the field of medical parasitology : the diagnosis of toxoplasmosis. However this study is weak and doesnt not add valuable data to the field.
Major comments:
What is the interest of such a study ? SAG1 has been used for decades (as stated by authors line 163). Thus why developping a new ELISA with this recombinant antigen ? what improvement in toxo diagnosis is expected by authors ?
The fact that LAT has weak performances is also known for decades
Why choosing 400 sera ? How were they selected ? On which criteria ? what is the profile of the patients ? How was the group of 80 sera defined ? Did the patients consent (not clear in the manuscript)
From a methodological point of view what is the serological reference to define a positive or negative serum? Is the commercial ELISA test published and what are its performances ?
Other comments:
Line 34: not only AIDS but also transplant patients are at risk of toxoplasmosis. Toxoplasma encephalitis is not acquired since in most of cases it is a reactivation of chronic reactivation.
Line 186: why is Madhurendra underlined ?
All along the text T. gondii should be in italics
References section: not up to date in the field
Author Response
Reviewer # 3:
This manuscript focuses on an important topic in the field of medical parasitology : the diagnosis of toxoplasmosis. However, this study is weak and doesn’t not add valuable data to the field.
Major comments:
What is the interest of such a study ? SAG1 has been used for decades (as stated by authors line 163). Thus, why developing a new ELISA with this recombinant antigen ? what improvement in toxo diagnosis is expected by authors ?
The fact that LAT has weak performances is also known for decades
Why choosing 400 sera ? How were they selected ? On which criteria ? what is the profile of the patients ? How was the group of 80 sera defined ? Did the patients consent (not clear in the manuscript)
From a methodological point of view what is the serological reference to define a positive or negative serum? Is the commercial ELISA test published and what are its performances ?
Other comments:
Line 34: not only AIDS but also transplant patients are at risk of toxoplasmosis. Toxoplasma encephalitis is not acquired since in most of cases it is a reactivation of chronic reactivation.
Line 186: why is Madhurendra underlined ?
All along the text T. gondii should be in italics
References section: not up to date in the field
Responses:
Major comments:
- What is the interest of such a study ? SAG1 has been used for decades (as stated by authors line 163). Thus, why developing a new ELISA with this recombinant antigen ? what improvement in toxo diagnosis is expected by authors ?
Basically, our study will be helpful for poor people, this kit will be local in Pakistan, three times cheap as compared to commercial Toxoplasma gondii ELISA kits and easily available at different labs, hospitals that is why Higher Education Commission of Pakistan provided the fund in the name of technology development fund (HEC-TDF03-239).
SAG1 has the features as compared to others are described below.
It is an immunodominant protein, stage specific (Toxoplasma Tachyzoital stage), good vaccine candidate. It is one of the most and highly immunodominant protein, increases with the reactivation of specific IgG antibodies from sera of patients having chronic toxoplasmosis.
It was described that SAG1 is prominent protein in manuscript (page number 8, line:174-176).
Native SAG1 protein has six intramolecular cysteine bridges that makes it immunodominant because of having immunologically confirmatory epitopes (39,40).
As described by Holec-Gąsior et al (2012)
“MIC1-MAG1-SAG1 Chimeric Protein, a Most Effective Antigen for Detection of Human Toxoplasmosis”
This new kit was developed on the local strain of toxoplasma of Pakistan, and it will be easily available, within short time available in the local labs, hospitals. It will be three time cheap as compared to Commercial Toxoplasma gondii ELISA kits.
what improvement in toxo diagnosis is expected by authors ?
The Human Toxo IgG ELISA kit has 100% specificity, and 98.5% sensitivity. It is highly accurate, fast, and cheap diagnostic kit. Short time (almost 90 min) required for test. The concentration of coating protein quantity is rSAG1 = 0.125 µg/ml that is very minute quantity as compared to other diagnostic kit.
- The fact that LAT has weak performances is also known for decades
LAT has weak performance, but in some article, it was showed that it is best confirmatory test and best test for diagnosis of toxoplasmosis.
Like Mazumder et al (1988) described that LAT has sensitivity of 84% and specificity of 100% in
“Latex agglutination test for detection of antibodies to Toxoplasma gondii. “
It is an initial screening test for diagnosis of toxoplasmosis (50) (Page no.9, line: 224), but not reliable and accurate method like rSAG1-based ELISA for the identification of toxoplasmosis (51) (Page no.9, line: 225).
Comment: Why choosing 400 sera ? How were they selected ? On which criteria ? what is the profile of the patients ? How was the group of 80 sera defined ? Did the patients consent (not clear in the manuscript)?
Response:
The 400 sera were chosen because we will find the seroprevalence of toxoplasmosis also. We have questionnaire for exclusion and inclusion criteria, so the samples were collected on these criteria. It is mentioned below.
Evaluation of indigenous ELISA kit based on recombinant surface antigen 1 for diagnosis and surveillance of toxoplasmosis in Humans and Halal Animals
Questionnaire related to Blood sampling
Sample. No…………………………………………………………….
Name:…………………………................................................
Age :……………………………………………………………………..
Sex:………………………………………………………………………
Region & Religion:…………………………………………………
District &Tehsil:……………………………………………………..
Medical Doctor/ Veterinary Doctor:…………………………….
Date:……………………………………………………………………….
The given information is required for the project and PhD study for identification of Toxoplasmosis and sampling method is done according to Random Sampling from humans of selected districts at Tehsil level and then at union council level.
Thanks for participation in this project.
- What is the source of drinking water if you consume at home………………….
- tape water nestle water c. commercial tape water
- Have you any cat in your house……………………….
- yes no
- Who handles the box of cat……………………………..
- myself other ones c. no one
- How can you clean your hands after handling cat-box? with…………………..
- simple water soap c. none
- Is there any pregnant woman in the house………………………………………
- yes no
- Is there any other pet animals in your house…………………………………….
- yes b. no
- If cat is present in your house, who puts the diet to cat........................................
- women b. men c. children
- What type of meat you offer to your cat…………………………………….
- cooked b. semi-cooked c. not offered
- Have you ever donated you blood………………………..
- yes b. no
9 (a) If yes, then how many times……………………………..
- 1 b. 2 c. many
9 (b) You even been diagnosed with a disease……………………..
- yes b. no
9 (c ) If yes then which ones………………………………………..
- Typhoid b. Toxoplasmosis c. Any other (………………….)
- What type of vegetables you use uncooked or as salad…………………………..
- spinach b. cauliflower c. any other (………………….)
- What type of milk you consume mostly……………………….
- unboiled b. boiled c. pasteurized d. none
Response: The 80 sera were selected on the following criteria to evaluate the performance of LAT (L), and Human Toxo IgG ELISA kit (K).
|
K+L+ |
29 Human sera |
|
K-L- |
25 Human sera |
|
K+L- |
26 Human sera |
Response: We collected human sera from Central Diagnostic Laboratory, Mayo Hospital, Lahore. It is mentioned in the manuscript (Page no.10, line: 255-256).
The consent form is described below.
Evaluation of indigenous ELISA kit based on recombinant surface antigen 1 for diagnosis and surveillance of toxoplasmosis in Humans and Halal Animals
Consent Form for Blood Sampling(Collection)
I, …………………… ……………, hereby solemnly declare that I have no objection for use of my blood for benefits of public in general and of pregnant women in particular. I especially allow my blood for diagnosis and awareness of Toxoplasmosis to the public benefits.
- I voluntarily participate in this good project Evaluation of indigenous ELISA kit based on recombinant surface antigen 1 for diagnosis and surveillance of toxoplasmosis in Humans and Halal Animals
- I show my interest to participate in this project for public benefits.
- I understand that even if I agree to participate now, I can withdraw at any time or refuse to answer any question without any consequences of any kind.
- I understand that I can withdraw permission to use data from my interview within two weeks after the interview, in which case the material will be deleted.
- I have had the purpose and nature of the study explained to me in writing and I have had the opportunity to ask questions about the study.
- I understand that I will not benefit directly from participating in this research.
- I understand that all information I provide for this study will be treated confidentially.
- I understand that in any report on the results of this research my identity will remain anonymous. This will be done by changing my name and disguising any details of my interview which may reveal my identity or the identity of people I speak about.
- I understand that if I inform the researcher that myself or someone else is at risk of harm they may have to report this to the relevant authorities - they will discuss this with me first but may be required to report with or without my permission.
- I understand that under freedom of information legalisation I am entitled to access the information I have provided at any time while it is in storage as specified above.
- I understand that I am free to contact any of the people involved in the research to seek further clarification and information.
Signature of research participant ----------------------------------------- date ----------------
I believe the participant is giving informed consent to participate in this study
Signature of researcher ---------------------------------------------- Date----------------------
Comment: From a methodological point of view what is the serological reference to define a positive or negative serum? Is the commercial ELISA test published and what are its performances ?
Response: The positive and negative Human sera were used in the commercial Toxoplasma gondii IgG ELISA kit that were obtained from WHO Code: TOX. A cut-off calibrator of 32 IU/ml was used based on this standardization. Catalogue number is KA0225 (ABNOVA company).
It has 98.3% sensitivity, and 99.2% specificity. The article for references of this commercial kit is described below:
- Turune, H.J., P.O. Leinikke, and K.M. Saari. Demonstration of intraocular Synthesis of Immunoglobin G Toxoplasma Antibodies for Specific diagnosis of Toxoplasmic Chorioretinitis by Enzyme Immunoassay. J. Clin. Microbiol. 17:988-992, 1983.
- Lin, T.M., S.P. Halbert and G.R. O’Connor. Standardized Quantitative Enzyme-linked Immunoassay for Antibodies to Toxoplasma Gondii. J. Clin. Microbiol. Vol. 11, 6:675-681, 1980.
- Roller, A., A. Bartlett and D.E. Bidwell. Enzyme Immunoassay with Special Reference ELISA Technique. J. Clin. Path. 31:507-520, 1987.
Other comments:
-Line 34: not only AIDS but also transplant patients are at risk of toxoplasmosis. Toxoplasma encephalitis is not acquired since in most of cases it is a reactivation of chronic reactivation.
Yes, I added such as AIDS and like other immunosuppressed diseases (page number 1 and line 34).
It is an immunocompromised disease so therefore it is the chances to reactivation of chronic reactivation of toxoplasmosis.
- Line 186: why is Madhurendra underlined: Madhurendra underlined word, the underline was mistakenly, now under line is removed, Madhurendra. (Page no.9 and line number 192).
- All along the text T. gondii should be in italics: The scientific name should be in italics. Now these are changed in the manuscript. (Page no.9, line:186, 227, 200, 206, 203, 201, 183, Page no.8, line:170,160,167,164,171, Page no.7, line:150, Page no.6, line:123).
- References section: not up to date in the field: All the references are up to dated according to the relevancy of the topic of the manuscript.
